# Individual-Level and Neighborhood-Level Factors Associated with Longitudinal Changes in Cardiometabolic Measures in Participants of a Clinic-Based Care Coordination Program: A Secondary Data Analysis

**DOI:** 10.3390/jcm11102897

**Published:** 2022-05-20

**Authors:** Sonal J. Patil, Mojgan Golzy, Angela Johnson, Yan Wang, Jerry C. Parker, Robert B. Saper, Debra Haire-Joshu, David R. Mehr, Randi E. Foraker, Robin L. Kruse

**Affiliations:** 1Department of Wellness and Preventive Medicine, Cleveland Clinic Community Care Institute, Cleveland, OH 44104, USA; saperr@ccf.org; 2Department of Family and Community Medicine, University of Missouri, Columbia, MO 65212, USA; ywang1201@gmail.com (Y.W.); mehrd@health.missouri.edu (D.R.M.); kruser@health.missouri.edu (R.L.K.); 3Biostatistics and Research Design Unit, School of Medicine, University of Missouri, Columbia, MO 65211, USA; golzym@health.missouri.edu; 4Center for Applied Research and Engagement Systems (CARES), University of Missouri, Columbia, MO 65211, USA; johnsonange@missouri.edu; 5Department of Physical Medicine and Rehabilitation, University of Missouri, Columbia, MO 65211, USA; parkerjc@missouri.edu; 6Brown School, Washington University in St. Louis, St. Louis, MO 63130, USA; djoshu@wustl.edu; 7Division of General Medical Sciences, School of Medicine, Washington University in St. Louis, St. Louis, MO 63130, USA; randi.foraker@wustl.edu

**Keywords:** care coordination, cardiometabolic risk, clinic-based intervention, nurse-led intervention, team-based care, neighborhood level factors

## Abstract

Background: Identifying individual and neighborhood-level factors associated with worsening cardiometabolic risks despite clinic-based care coordination may help identify candidates for supplementary team-based care. Methods: Secondary data analysis of data from a two-year nurse-led care coordination program cohort of Medicare, Medicaid, dual-eligible adults, **L**everaging **I**nformation Technology to **G**uide **H**igh **T**ech, **H**igh **T**ouch Care (LIGHT^2^), from ten Midwestern primary care clinics in the U.S. *Outcome Measures*: Hemoglobin A1C, low-density-lipoprotein (LDL) cholesterol, and blood pressure. Multivariable generalized linear regression models assessed individual and neighborhood-level factors associated with changes in outcome measures from before to after completion of the LIGHT^2^ program. Results: 6378 participants had pre-and post-intervention levels reported for at least one outcome measure. In adjusted models, higher pre-intervention cardiometabolic measures were associated with worsening of all cardiometabolic measures. Women had worsening LDL-cholesterol compared with men. Women with pre-intervention HbA1c > 6.8% and systolic blood pressure > 131 mm of Hg had worse post-intervention HbA1c and systolic blood pressure compared with men. Adding individual’s neighborhood-level risks did not change effect sizes significantly. Conclusions: Increased cardiometabolic risks and gender were associated with worsening cardiometabolic outcomes. Understanding unresolved gender-specific needs and preferences of patients with increased cardiometabolic risks may aid in tailoring clinic-community-linked care planning.

## 1. Introduction

Cardiovascular disease (CVD) is the leading cause of death in the U.S., and cardiometabolic risk factors for CVD are prevalent and well-known [1,2]. Despite the availability of effective treatments, significant disparities in CVD and cardiometabolic outcomes persist [3,4,5,6]. Care coordination can reduce fragmentation in care, and team-based care with community-based support can help address behavioral and social determinants of cardiometabolic risks [7,8,9,10,11,12]. However, most primary care practices do not have the time or resources to offer widespread care coordination, individual-level social and behavior risk screenings, and team-based care [13,14,15,16,17,18]. Tailoring team-based approaches to high-burden communities, people living in poverty, or those with low literacy is recommended for reducing CVD disparities [11]. Support outside busy clinic workflows needs to be tailored to individuals who do not benefit from clinic-based interventions. Advances in geospatial technologies have increased electronic health record (EHR) access to community-level geocoded data for social and behavioral risks associated with each patient’s residential address [19]. Identifying clinical, sociodemographic, and community-level factors that predict an individual’s health outcomes after participating in clinic-based interventions can help identify potential candidates for supplementary clinic-community linked interventions [20,21]. We found no studies examining patient-level moderating factors of cardiometabolic outcomes with nurse-led care coordination. Hence, we aimed to identify patients’ clinical, sociodemographic, and neighborhood-level factors associated with less improvement or worsening cardiometabolic outcomes despite participation in a 2-year clinic-based nurse-led care coordination program.

## 2. Materials and Methods

We performed a secondary analysis of data from a prospective cohort of University of Missouri Healthcare (MUHC) system patients enrolled in the **L**everaging **I**nformation Technology to **G**uide **H**igh **T**ech, **H**igh **T**ouch Care (LIGHT^2^) project from 1 July 2013 to 30 June 2015 [22]. The University of Missouri institutional review board determined the LIGHT^2^ program to be a quality improvement activity not requiring institutional review board review. We used the Strengthening the Reporting of Observational Studies in Epidemiology (STROBE) reporting guidelines for this study [23]. 

### 2.1. Study Design and Setting

Funded with a Centers for Medicare and Medicaid Services (CMS) innovation award, the LIGHT^2^ program was a combination of information technology components (High Tech) and health care coordination by nurse care managers (High Touch) [22] (pp. 231–292), [24]. The High-Tech component included dashboards and a patient portal for communication with physicians and nurse care managers. The High Touch component was care coordination provided by nurse care managers; 25 nurse care managers worked in 10 family and community medicine and internal medicine clinics providing services. Patients were assigned tiers based on their chronic disease diagnoses and health care utilization in the previous 12 months [25]. Nurse care managers provided as-needed education and support for patients regarding new chronic disease diagnoses or worsening of chronic conditions. A documentation system measuring the frequency and time of nurse care manager contacts was created using the Agency of Healthcare Research and Quality (AHRQ) care coordination atlas [24,26]. The study compiled data on claims and participants’ EHR data, including healthcare utilization, clinic visits, basic demographic data, diagnosis, labs, and nurse care coordination contacts, along with geocoded patient addresses. Cohort recruitment started in February 2013; 9932 participants were recruited by July 2013. Deaths and relocations decreased the number of participants to 8593 by March 2015. The goals of the LIGHT^2^ intervention were to achieve net cost savings, increase preventive service use, and improve the management of chronic diseases. The final evaluation of the program showed higher spending and inpatient admissions than the comparison group [22] (pp. 231–292). Nominal improvements were seen for chronic disease management across all tiers. In higher-risk tiers, patients’ Hba1c control improved from 18% to 12% but worsened slightly in low-risk tiered patients from 9% to 10%. The LDL cholesterol did not change in the high-risk tiers, whereas for patients in low-risk tiers, LDL cholesterol controls improved from 60% to 71%. Hypertension control worsened in low-risk tiered patients by 23% and in high-risk tiered patients by 29% [22]. The cardiometabolic measures monitored during the LIGHT^2^ intervention were hemoglobin A1C (HbA1C), blood pressure (B.P.), and low density-lipoprotein (LDL) cholesterol. Primary results showed minimal changes in cardiometabolic outcomes. Detailed results of primary outcomes, nurse care coordination implementation and fidelity of LIGHT^2^ have been previously published [22] (pp. 231–292), [24].

### 2.2. Participants

The participants were all Medicare, Medicaid, or dual-eligible patients receiving primary care services in any of 10 family and community medicine or internal medicine clinics in the MUHC system during the project period. Because we examined factors associated with a change in cardiometabolic measures, we included only participants with at least one of the three outcomes, HbA1C, LDL cholesterol, or B.P. reported before and after the intervention completion.

### 2.3. Outcomes

We extracted HbA1C, BP, and LDL-cholesterol levels before and after the LIGHT^2^ program. The recruitment was complete, and the care coordination documentation system was implemented on 1 July 2013. We used measures reported within 6 months before that date or before the first nurse care manager contact within the first 3 months of the intervention as the pre-intervention values. For post-intervention measures, we used the values reported within 6 months after the intervention completion date. If there were no values reported in the 6 months after the completion of the intervention, we used the last reported value within the last 3 months of the intervention. We used the average of the last two reported B.P. readings if there were multiple readings in the defined pre-intervention and post-intervention time-period. Individuals with very low pre-intervention values were not included in the analyses. More specifically, if pre-intervention LDL < 60 mg/dL, pre-intervention HbA1C < 5.5%, pre-intervention diastolic B.P. < 40 mm of Hg, and pre-intervention systolic B.P. < 90 mm of Hg, individuals were excluded from the respective analysis. We also excluded individuals with two extremely high values (above 350) of post-intervention LDL cholesterol from the analysis for LDL-cholesterol outcomes. The population was comprised of 6378 individuals after exclusions. 

### 2.4. Variables

*Factors that affect the effectiveness of health interventions:* We included clinical and geospatial variables accounting for the majority of the National Academy of Medicine (NAM) recommended social and behavioral domains and measures that influence health outcomes and effectiveness of treatments [27,28]. We identified clinical, sociodemographic, and area-level proxy variables that can be easily extracted from routinely collected clinical and sociodemographic data in the EHR. The variables included for NAM recommended domains were: (1) race and ethnicity, which was extracted at the individual-level from EHR; (2) education, which is also a proxy for health literacy, was extracted at the neighborhood level; (3) financial resource strain was extracted as poverty and access to healthy food at the neighborhood level; (4) stress and depression were denoted by the presence of mood disorders at an individual level from EHR; (5) physical activity opportunities were extracted at neighborhood level; (6) tobacco and alcohol status was extracted at individual level from EHR; (7) social connections and isolation were measured by marital status at individual level in EHR and neighborhood-level social capital; and (8) intimate partner violence was measured by neighborhood-level domestic violence injury rates. 

Clinical and sociodemographic variables: We included basic demographic data such as age, sex, and race/ethnicity. We extracted smoking and alcohol use status as of the first day of the intervention. We extracted the total number of comorbidities, nurse care manager contacts, and presence or absence of a mood disorder reported as of the last day of the intervention based on the International Classification of Diseases, Ninth Revision, Clinical Modification diagnosis codes (ICD 9 codes). As baseline cardiometabolic measures, including body mass index (BMI), are used by clinicians for CVD risk stratification, we adjusted for baseline values of cardiometabolic measures. As the LIGHT2 program was focused on nurse-led care coordination for healthcare utilization and chronic disease management, we adjusted for health resource utilization, nurse care manager contacts, and the number of comorbidities. We defined high-resource healthcare utilizers as individuals with four or more emergency room visits or two or more inpatient encounters in the previous 12 months.

Geospatial variables: The patient’s residential address and primary care physician’s (PCP) clinic addresses were geocoded using ArcGIS Online World Geocoding Service [29]. We included neighborhood-level measures of poverty, education, social capital, domestic violence injury rates, access to healthy food and physical activity, and driving distance to primary care clinics. If a patient was seen at two or more primary care clinic locations during the intervention period, we coded the distance to the clinic as missing to avoid misattributions. For calculating the travel distance to the closest grocery store from patients’ geocoded address, supermarket locations were obtained from the Reference USA US Businesses dataset [30,31]. The neighborhood-level poverty was extracted using the percentage of population <200% federal poverty level (FPL) and was extracted at the census-tract level, and the percentage of the population without high a school diploma was determined at the census-block group level [32]. Neighborhood social capital for patients’ ZIP-code level was assessed using the number of civic or social organizations per capita, obtained by summarizing data from the 2017 U.S. Census Bureau ZIP code Business Patterns [33]. Domestic violence injury rates were extracted at the ZIP-code level of the patient’s geocoded address. To assess opportunities for physical activity, we generated WalkScore™ values for each patient’s census block [34]. The visual assessment found that the WalkScore™ dataset did not represent physical activity opportunities in areas with networks of unpaved trails, such as our local county and rural areas; hence, we excluded these measures. See Appendix A for a detailed description of methods used for determining neighborhood-level social risks in our analysis.

### 2.5. Statistical Methods

After adjusting for covariates, we examined post-intervention HbA1C, LDL, and B.P. controlling for pre-intervention values (from before the 1 July 2013 nurse-led care coordination start date to after the end of the project in June 2015). A dataset was created with de-identified records for each patient containing their clinical and sociodemographic information and geocoded residential addresses. Geocoded residential addresses of patients were used to extract neighborhood-level variables. The data analysis was conducted on data between November 2019 and October 2020. We fitted a separate generalized linear model (GLM) for each of these measures as the dependent variable. We adjusted for covariates, including pre-intervention measures and pre-intervention body mass index (BMI). The GLM model assumptions were checked. Using the Pearson correlation between continuous variables, we chose one variable from each group of variables. Variance Inflation Factors (VIF) were calculated as 1/(type I tolerance) of the GLM model for categorical variables, and a cutoff point of 5 was used. The overall fit of the models was assessed using the coefficient of determinations (R^2^). The R^2^ value shows the percentage of variance in the dependent variable that is predictable from the independent variables collectively and gives the strength of the relationship. Although there is no standard for acceptable R^2^, it is suggested that the R^2^ values of 0.02, 0.13, 0.26 correspond to small, medium, and large effect sizes, respectively [35]. We report statistical significance at *p* < 0.05. The analysis was performed using SAS software. 

*Model 1:* Included covariates readily available in the clinical chart as follows: age, gender, race/ethnicity, marital status, smoking status, alcohol use, healthcare utilization 12 months before the study’s start date, pre-intervention cardiometabolic measure levels, pre-intervention BMI, presence of mood disorder, total number of chronic health conditions, and total number of nurse care manager contacts during the study period. Significant interactions between the covariates and the pre-intervention values were included in the model. 

*Model 2:* Included the covariates in Model 1 and also included neighborhood-level measures of poverty, education, social capital, and domestic violence injury rates. 

*Model 3:* Included the covariates in Model 2 and added physical determinants of access to healthy food (grocery store) and healthcare (PCP clinic location) from the patient’s geocoded address. 

*Parsimonious Model:* We used a backward-selection technique, with a level of significance of 0.05 to obtain a parsimonious model. Parsimonious models achieve better predictability and desired level of goodness of fit with as few explanatory variables as possible.

At the census-block group level, we noted collinearity among the percentage of the population below 200% of the FPL and the Area Deprivation Index [36]. Thus, we included only the percentage of the population below 200% of the FPL in our models. We dichotomized driving time to PCP clinic location to 30 min or less and over 30 min, as Health Professional Shortage Area (HPSA) designation utilizes driving time of more than 30 to 40 min to the nearest source of care as one of the scoring criteria [37]. Based on frequency distributions and outliers, we represented the percentage of the population without a high school diploma and the number of nurse care manager contacts as categorical variables by quartiles. 

*Sensitivity analysis:* We performed two additional analyses for each outcome: (1) We fitted a mixed-effects model to test for random variability across clinics. (2) We fitted a mixed-effects model to test for random variability across all included patients’ census tracts (137 census tracts). Various area-level measures included in our analyses were associated with different geographic areas; hence, an overall nested analysis was not possible. 

*Power analysis:* There were 8593 participants in the LIGHT^2^ cohort. With 6000 participants, we were well-powered to detect clinically important differences in CVD risk factors. For example, we will be able to detect a 4 mm difference in systolic blood pressure at the 0.05 level of significance with greater than 85% power.

## 3. Results

Of 8593 eligible participants in the LIGHT^2^ cohort, 6378 participants had at least one cardiometabolic measure reported both before and after the intervention. The Figure 1 Flow Diagram illustrates the derivation of participants for each cardiometabolic measure. 

The cohort description is included in Table 1. Because our cohort was primarily White, non-Hispanic ethnicity (86.3%), we dichotomized the race/ethnicity variable. The presence of mood disorder was reported in 1475 (23.13%) participants, of which 32.36% were male participants and 67.64% were female participants.

We obtained LDL cholesterol outcomes for 2377 participants, HbA1C for 1290 participants, and B.P. outcomes for 4619 participants. The results of all models were consistent. The final parsimonious models showed significant worsening of LDL-cholesterol associated with higher pre-intervention LDL-cholesterol levels (β 0.56, 95% CI 0.52 to 0.60, *p* < 0.001), significant worsening of HbA1C associated with higher pre-intervention HbA1C levels (main effect β 0.51, 95% CI 0.43 to 0.59, *p* < 0.001), significant worsening of systolic and diastolic BP with higher pre-intervention systolic BP (main effect β 0.95, 95% CI 0.83 to 1.08, *p* < 0.001), and higher pre-intervention diastolic BP (main effect β 0.83, 95% CI 0.75 to 0.91, *p* < 0.001), respectively. LDL-cholesterol worsened for women compared with men (β 7.76, 95% CI 5.21 to 10.32, *p* < 0.001). There was a significant interaction between pre-intervention HbA1C and gender (main effect β −1.29, 95% CI −1.95 to −0.62, *p* < 0.001; interaction effect β 0.19, 95% CI 0.09 to 0.28, *p* < 0.001), with HbA1C worsening if the pre-intervention HbA1C was more than 6.8% in women compared with men (Appendix A). The interaction between gender and pre-intervention systolic BP levels (main effect β −7.86, 95% CI −15.55 to −0.17 *p* = 0.045; interaction effect β 0.06, 95% CI 0.002 to 0.12, *p* = 0.043) indicates worsening trend in systolic BP for women compared with men with pre-intervention systolic BP level >131 mm Hg. All other associations were inconsistent across all cardiometabolic measures. The coefficient of determinations (R^2^) did not change significantly with the addition of neighborhood-level variables. See Table 2, Table 3 and Table 4 for results from parsimonious models for LDL- cholesterol, HbA1C, and systolic B.P. outcomes and Appendix A for results from the parsimonious model for diastolic B.P. outcomes. See Appendix A for detailed models for all measures and R^2^ for all models. See Appendix A for the interaction of female sex with pre-intervention systolic B.P. on adjusted change in systolic B.P. and diastolic B.P., respectively. See Appendix A for the interaction of female sex with pre-intervention HbA1C on adjusted change in HbA1C. The interaction between female sex and mood disorder was only significant for diastolic BP outcomes (main effect β −5.33, 95% CI −9.75 to −0.19 *p* = 0.018; interaction effect β 1.63, 95% CI 0.55 to 2.98, *p* = 0.003). The results from both the mixed-effects model for clinics and census tracts show no significant changes in the parameter estimates for significant associations. Appendix A give the solution to the random effect of clinics and the variability.

## 4. Discussion

In predominantly white suburban and rural participants from a two-year nurse-led care coordination program, we found higher pre-intervention cardiometabolic measures were associated with worsening post-intervention levels. Individuals with increased cardiometabolic risks may not benefit from solely clinic-based care coordination and may need additional support for self-management between clinic visits. Additionally, we found women’s LDL-cholesterol worsened compared with men, and women with high pre-intervention HbA1C (>6.8%) and B.P. (>131 mm Hg systolic B.P.) became worse compared with men despite clinic-based, nurse-led care coordination. In women with increased cardiometabolic risks, there may be additional psychosocial or behavioral contexts that may not be addressed by solely clinic-based care coordination. While there were some inconsistent findings across outcomes, the addition of neighborhood-level risks based on geocoded residential addresses did not significantly change the associations or fit of models compared to models based on variables that are routinely available in clinical charts. This is consistent with previous studies investigating the impact of community-level determinants on risk predictions and outcomes [21,38,39]. There may be additional gender-specific social and behavioral contexts that impact cardiometabolic risks that are not captured by EHR or neighborhood-level variables.

Our results differ from several previous studies that have shown nurse case management improves diabetes and hypertension control [40,41]. Our results may be different from previous studies as nurses in the LIGHT^2^ program provided care coordination and as-needed chronic disease education during clinic visits. Most other studies of nurse-led interventions included additional case management or disease management components delivered during and between clinic visits [40,41,42]. Individuals with higher baseline cardiometabolic risks may have unresolved needs and preferences not addressed by solely clinic-based care coordination. Gender is the only socially stratifying factor present in >50% of our cohort [43]. Our findings of worsening cardiometabolic outcomes in women compared with men when baseline B.P. and HbA1c were high may indicate additional psychosocial contexts for women with uncontrolled hypertension or diabetes that limit their ability to benefit from solely clinic-based interventions [44,45,46,47]. Further research to understand and address gender-specific needs and preferences in individuals with increased cardiometabolic risks may help tailor team-based care between the clinic and community-based team members.

*Strengths of our study:* Our study is the first study assessing the interaction of clinical factors and neighborhood-level social risks to identify individuals that may not benefit from clinic-based care coordination. We only included variables that can be extracted from clinical charts or based on the patient’s home address in the demographic section of clinical charts to avoid burdening PCPs with additional individual social and behavioral risk screenings [15,16]. We separated our models based on covariates available in clinical charts, covariates extracted at the neighborhood-level (zip code or census tract), and covariates extracted by calculating the distance from geocoded addresses. In addition to variables accounting for the NAM recommended social and behavioral domains, our study included one factor associated with each of the five key areas of social determinants of health included in the place-based organizing framework developed by Healthy People 2020; namely, economic stability, education, social and community context, neighborhood environment, and healthcare [48,49].

*Weaknesses of our Study:* We acknowledge several limitations. First, the cohort consists of Medicaid, Medicare, or dual-eligible beneficiaries of predominantly white, non-Hispanic ethnicity from Midwestern primary care clinics, which may limit the generalizability of our findings. Second, physical activity is one of the primary CVD risk factors, but we could not identify reliable neighborhood-level measures of physical activity opportunities for our cohort of suburban and rural participants [50,51]. Third, though burdensome, individual-level social and behavior risks, rather than neighborhood-level risks, can improve the ability to predict which patients may benefit from supplementary clinic-community linked interventions [38,52,53]. Lastly, we acknowledge that the LIGHT^2^ care coordination program focused on super-utilizers and reducing care fragmentation rather than reducing health disparities [22].

## 5. Conclusions

We found higher baseline cardiometabolic risks and gender were commonly associated with worsening cardiometabolic outcomes in predominantly White suburban and rural participants from a clinic-based care coordination program. The addition of neighborhood-level risks based on patients’ residential addresses did not change estimates of associations beyond routinely collected clinical and sociodemographic data in EHR. Further research to understand gender-specific needs and preferences of individuals with increased cardiovascular risks may aid in tailoring clinic-community-linked team-based care.

## Figures and Tables

**Figure 1 jcm-11-02897-f001:**
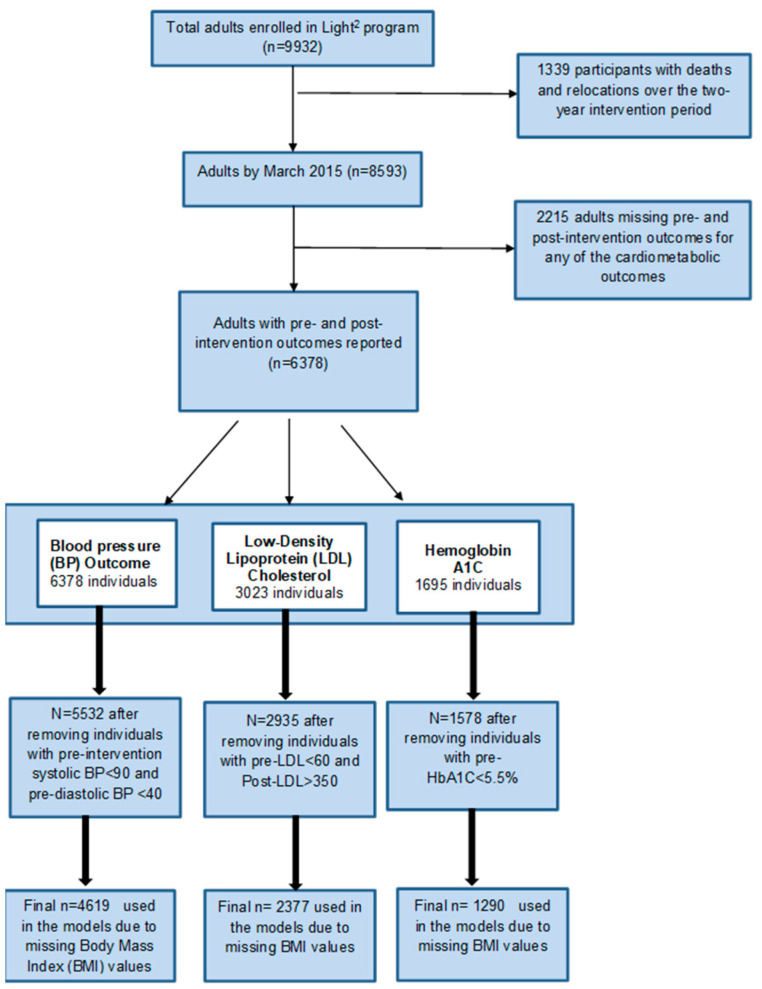
Flow Diagram illustrating the derivation of participants for each cardiometabolic measure.

**Table 1 jcm-11-02897-t001:** Characteristics of the study sample (N = 6378).

Variable	Total
Demographics	
Age (mean, SD)	62.67 (18.5)
Sex (frequency [%])	
Female	3928 (61.59)
Male	2450 (38.41)
Race (frequency [%])	
White non-Hispanic	5507 (86.34)
Other	871 (13.66)
Marital status (frequency [%])	
Married	2746 (43.05)
Other	892 (13.99)
Single	1622 (25.43)
Widowed	1118 (17.53)
Cardiometabolic measures (mean, S.D.)	
Pre-intervention LDL (mg/dL)	106.26 (31.55)
Post-intervention LDL (mg/dL)	99.41 (36.22)
Pre-intervention HbA1c (%)	6.94 (1.43)
Post-intervention HbA1c (%)	7.09 (1.53)
Pre-intervention systolic BP (mm of Hg)	132.74 (14.17)
Post-intervention systolic BP (mm of Hg)	131.14 (17.61)
Pre-intervention diastolic BP (mm of Hg)	75.83 (7.64)
Post-intervention diastolic BP (mm of Hg)	75.05 (9.69)
Neighborhood characteristics	
Percentage of population below 200% of the FPL (mean, SD) for patient’s census-tract	36.14 (14.30)
Percentage of population that did not graduate from high school for patient’s census-block group (frequency [%])	
Quartile 1: <3.17%	2049 (32.13)
Quartile 2: 3.17–8.79%	1519 (23.82)
Quartile 3: 8.79–14.06%	1325 (20.77)
Quartile 4: >14.06%	1485 (23.28)
Domestic violence injury rates (per 1000) for patient zip codes, 2011–2015 (mean, SD)	0.39 (0.37)
Distance to nearest grocery store from patient’s geocoded address (miles)	3.42 (3.69)
Number of civic or social organizations per capita for patient zip codes (mean, SD)	13.10 (4.9)
Health characteristics	
Number of comorbidities	4.89 (4.23)
Pre-intervention body mass index (kg/m^2^)	30.35 (7.71)
Current smoking (frequency [%])	
Yes	1455 (22.81)
No	4810 (75.42)
Missing	113 (1.77)
High-risk alcohol use (frequency [%])	
Yes	110 (1.72)
Unknown	6368 (98.28)
Presence of mood disorder (frequency [%])	
Yes	1475 (23.13)
Unknown	4903 (76.87)
Total number of nurse care manager contacts during the study period (frequency [%])	
Quartile 1: <5	1504 (23.58)
Quartile 2: 5–10	1489 (23.35)
Quartile 3: 11–21	1745 (27.36)
Quartile 4: >22	1640 (25.71)
High versus low healthcare resource utilizer (frequency [%])	
High utilizer	778 (12.2)
Low utilizer	5600 (87.8)
Travel time to PCP office from geocoded addresses (frequency [%])	
<=30 min	4004 (62.78)
>30 min	1331 (20.87)
Unknown	1043 (16.35)

SD = standard deviation, LDL = low density lipoprotein cholesterol, HbA1c = glycosylated hemoglobin, BP = blood pressure, FPL = federal poverty level, PCP = primary care provider.

**Table 2 jcm-11-02897-t002:** Results of Parsimonious model for LDL cholesterol outcome (R^2^ = 0.29).

Parameter	Adjusted β (95% Confidence Limits)	*p*-Value
Intercept	68.66 (57.87, 79.44)	<0.001
Pre-intervention BMI	−0.19 (−0.35, −0.02)	0.02
Pre-intervention LDL	0.56 (0.52, 0.60)	<0.001
Female (ref = male)	7.76 (5.21, 10.32)	<0.001
Non-White race (ref = White)	−3.43 (−7.24, 0.38)	0.077
Age	−0.26 (−0.36, −0.17)	<0.001
Number of comorbidities	−0.47 (−0.79, −0.15)	0.004
Percentage of area population below 200% of the FPL	−0.14 (-0.23, −0.05)	0.002
Domestic violence injury hospitalization rate (per 1000 population)	−5.78 (−9.24, −2.33)	0.001

LDL = low density lipoprotein, BMI = body mass index (kg/m^2^), ref = reference category, FPL = federal poverty level, R^2^ = unadjusted coefficient of determinations.

**Table 3 jcm-11-02897-t003:** Results of Parsimonious model for HbA1C outcome (R^2^ = 0.39).

Parameter	Adjusted β (95% Confidence Limits)	*p*-Value
Intercept	3.73 (2.93, 4.53)	<0.001
Pre-intervention HbA1C	0.51 (0.43, 0.59)	<0.001
Female (ref = male)	−1.29 (−1.95, −0.62)	<0.001
Pre-intervention HbA1C × female sex	0.19 (0.09, 0.28)	<0.001
Non-White race (ref = White)	−1.16 (−1.94, −0.37)	0.004
Pre-intervention HbA1C × non-White race	0.14 (0.03, 0.25)	0.01
Age	−0.006 (−0.012, −0.00001)	0.05
Current smoker (ref = No)		
Yes	−0.20 (−0.38, −0.017)	0.03
Unknown	−0.58 (−1.16, −0003)	0.05
Civic and social associations rate (per 100,000 population)	0.01 (−0.0008, 0.026)	0.06
Distance to nearest grocery store from patient’s geocoded address in miles	0.01 (−0.007, 0.03)	0.25

HbA1C = hemoglobin A1C; ref = reference category, × = interaction symbol, R^2^ = unadjusted coefficient of determinations).

**Table 4 jcm-11-02897-t004:** Results of Parsimonious model for Systolic B.P. (R^2^ = 0.43).

Parameter	Adjusted β (95% Confidence Limits)	*p*-Value
Intercept	−3.68 (−19.70, 12.34)	0.65
Pre-intervention BMI	0.096 (0.04, 0.15)	<0.001
Pre-intervention SBP	0.95 (0.83, 1.08)	<0.001
Female (ref = Male)	−7.86 (−15.55, −0.17)	0.045
Pre-intervention SBP × female sex	0.06 (0.002, 0.12)	0.043
Age	0.38 (0.17, 0.60)	<0.001
Pre-intervention SBP × age	−0.003 (−0.004, −0.0008)	0.003
Number of comorbidities	0.10 (0.009, 0.19)	0.03
Female sex × mood disorder	1.81 (0.96, 1.9)	0.058
Percentage of area population below 200% of the FPL	0.30 (0.18, 0.43)	<0.001
Pre-intervention DBP × Percentage of area population below 200% of the FPL	−0.004 (−005, −0.002)	<0.001
Domestic violence injury hospitalization rate (per 1000)	2.21 (1.16, 3.26)	<0.001

BMI = body mass index (kg/m^2^), SBP = systolic blood pressure, FPL = Federal poverty level, DBP = diastolic blood pressure, × = interaction symbol, R^2^ = unadjusted coefficient of determinations.

## Data Availability

The LIGHT^2^ project was funded by a Centers for Medicare and Medicaid Services (CMS) innovation award and all data was shared with the funding agency [https://downloads.cms.gov/files/cmmi/hcia-crppm-thirdannrptaddendum.pdf] (pp. 231–292) (accessed on 10 May 2022). The datasets used and/or analyzed during the current study are available from the corresponding author on reasonable request.

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
