# Peer review of "Individual-Level and Neighborhood-Level Factors Associated with Longitudinal Changes in Cardiometabolic Measures in Participants of a Clinic-Based Care Coordination Program: A Secondary Data Analysis"

_jcm, 2022, doi:10.3390/jcm11102897_

Round 1

Reviewer 1 Report

It has been very interesting for me to read the work of Patil SJ et al., it deals with a subject of great current interest and both the objectives and the methodology, the results and its conclusions must be shown to the interested scientific community. 

This paper presented by the authors raises the question of identifying clinical, sociodemographic, and patient neighborhood-level factors associated with less improvement or worsening of cardiometabolic outcomes despite participation in a nurse-led care coordination program in a clinic 2 years.

This is a relevant topic since it addresses the important role that nursing has in good control and achievement of cardiometabolic goals in the community of the health care population.

This is an issue that, although previously discussed, needs a greater approach and awareness on the part of health teams, especially in primary care.
The authors analyze a cohort of more than 8,500 patients and propose a very adequate methodology, the results are consistent. In addition, it is a well-written and easy-to-read study, with consistent conclusions, identifying the profile of the patient (by gender) with the highest cardiometabolic risk.

Author Response

We are grateful for your review and feedback.

Thank you,

Regards,

Sonal Patil

Reviewer 2 Report

This is a huge study of nurse-led care coordination program. Followed variables were HbA1C, BP and LDL. The most important finding was that this mostly educational and coordinational program was not efficient in those patients whose risk factors were worse in pre-intervetion study.

The problem of this manuscript is the huge number of parameters described in methods. It is hard to find to major points from this information. The reference to find the information of LIGHT2 (ref nro 22) is raport that includes lot of information.

I do not know the parsimonious model well enough. In backward -selecion is p<0,10 the limit of decision making? How r2 should be interpreted? Please write statistical methods so that references are not needed for interpretation.

Model 1,2 and 3 are presented only in supplemental material are those needed in the manuscript?

Author Response

We are grateful for the reviewers’ comments and suggestions, which helped make our manuscript simpler and easier to follow. We are pleased to enclose the attached letter with a response to reviewer comments on our manuscript "Individual-level and neighborhood-level factors associated with longitudinal changes in cardiometabolic measures in participants of a care-coordination program."

Our responses are in blue. Excerpts from the revised manuscript are in quotation marks.

We truly appreciate your considering the revision of our manuscript. Please do not hesitate to contact me if I can provide any further information.

Thank you,

Sincerely,

Sonal J. Patil, MD, MSPH

Staff, Clinician-Investigator

Department of Wellness and Preventive Medicine, Cleveland Clinic Community Care

1950 Richmond Road, Lyndhurst, OH 44124

Round 2

Reviewer 2 Report

Thank you for revised version. I think revised manuscript is now easier for reader. No more suggestion or comments.

Author Response

We are grateful for your time and review.